

# A new origin of the 'modern' lungfish dentition revealed by taxonomic overlap between Devonian and Carboniferous dipnoans

Amin El Fassi El Fehri[1], Alice M. Clement[2], Jorge Mondéjar Fernández[3], Merle Greif[1] and Christian Klug[1]

[1] Department of Paleontology, University of Zürich, Zurich, Switzerland
[2] College of Science and Engineering, Flinders University, Adelaide, South Australia, Australia
[3] Division of Paleontology and Historical Geology, Senckenberg Research Institute and Natural History Museum, Frankfurt am Main, Germany

Corresponding author
Amin El Fassi El Fehri,
amin.elfassielfehri@pim.uzh.ch

## ABSTRACT

Lungfishes (Dipnoi, Sarcopterygii) initially radiated in the Early Devonian, and reached the apogee of their diversity during this period, especially with regard to their dentitions. Following the end-Devonian extinction, most of this diversity was lost and remained low throughout the Carboniferous and the rest of the Palaeozoic, mainly represented by the incredibly successful '*Sagenodus*-like' dental morphology with sharp rows of fused teeth. Nevertheless, the exact scenario of lungfish evolution across the Devonian-Carboniferous boundary remains ambiguous. Recent work on new dipnoan assemblages from the Famennian (Upper Devonian) and Tournaisian (Lower Carboniferous) has challenged our understanding of lungfish evolution across this boundary. These studies suggest that the end-Devonian extinction did not impact lungfishes as strongly as other sarcopterygians, and that many Carboniferous lineages have origins that stretch much further back in time. However, concrete fossil evidence supporting these new hypotheses remained exceedingly rare. Here, we describe a highly derived lungfish tooth plate from the Famennian of the Tafilalt region in Morocco. This specimen's morphology is akin to that of the Carboniferous genus *Sagenodus*, often dubbed as the earliest 'modern' lungfish. Although the material is not sufficient for a precise taxonomic identification or placement with phylogenetic analyses, it shows that a Carboniferous lineage—the Sagenodontidae—extends deep roots into the Devonian. This specimen supports recently developed ideas about lungfish evolution blurring across the Devonian-Carboniferous boundary and pushes back the origin of the 'modern' lungfish dental morphotype by some 20 million years from the Visean to the Famennian.

## INTRODUCTION

Lungfishes (Dipnoi) form one of two extant lineages of lobe-finned fishes (Sarcopterygii), excluding tetrapods (*Schaeffer, 1952*; *Ahlberg, 1991*; *Schultze, 1994*; *Lu et al., 2017*; *Cui et*

*al., 2022*). Today, they are only represented by six species, but the group once reached incredible diversity after its initial radiation at the beginning of the Devonian (*Lloyd, Wang & Brusatte, 2012*; *Brazeau & Friedman, 2015*; *Challands et al., 2019*). Our understanding of dipnoan evolution after this period of taxonomical and morphological richness, and across the end-Devonian mass extinction (Hangenberg event) remains unclear. To explore lungfish evolutionary trends during this time, their dentitions provide excellent study material as they are robust, compact, and highly mineralised elements, which fossilise often and well (*Owen, 1867*; *Kemp, 1977*; *Campbell & Barwick, 1984*; *Bemis, Burggren & Kemp, 1987*; *Smith & Chang, 1990*; *Ahlberg, Smith & Johanson, 2006*; *Mondéjar-Fernández et al., 2020*). Although not the best indicators of phylogenetic affinity (*Reed, 1985*; *Kemp, 1997*; *Kemp, 2005*; *Mondéjar-Fernández et al., 2020*), their abundance and preservation potential can give us great insights into changes in disparity and diversity through deep time.

Devonian lungfish dentitions are remarkably disparate. They broadly fall into three categories: tooth plates, dentine plates, and denticulated plates, as well as some intermediate forms (*Campbell & Barwick, 1983*; *Campbell & Barwick, 1990*; *Ahlberg, Smith & Johanson, 2006*; *Campbell & Barwick, 2008*; *Mondéjar-Fernández et al., 2020*). This classification is not phylogenetically valid (*Schultze & Marshall, 1993*; *Ahlberg, Smith & Johanson, 2006*), but it is representative of the dental disparity in Devonian dipnoans. Dentine plates composed of thick sheets of dentine are only known in the Devonian (*e.g., Chirodipterus* and *Dipnorhynchus*) (*Miles, 1977*; *Campbell & Barwick, 2000*), and denticulated plates ornamented with small denticles occur in both the Devonian and earliest Carboniferous (*e.g., Uranolophus, Melanognathus, Conchopoma*) (*Campbell & Barwick, 1988*; *Smith & Krupina, 2001*; *Ahlberg, Smith & Johanson, 2006*). Tooth plates on the other hand, represent the ancestral and most successful form of lungfish dentition, and the only one that persisted into post-Palaeozoic times (*Reisz & Smith, 2001*; *Ahlberg, Smith & Johanson, 2006*). Devonian tooth plates have individual, rounded tooth cusps, partially organised in rows as seen in the Early Devonian taxa *Diabolepis speratus* and *Tarachomylax oepiki* (*Chang & Yu, 1984*; *Barwick, Campbell & Mark-Kurik, 1997*). This arrangement in rows appears to become more stringent in later taxa such as *Dipterus valenciennesi* and *Andreyevichthys epitomus* (*Krupina, 1987*; *Long, 1987*; *den Blaauwen, Barwick & Campbell, 2006*). The overall post-Devonian fossil record then shows a general shift toward more similar morphotypes. Early Carboniferous (Mississippian) tooth plates display partially fused tooth cusps that form compact rows (*e.g., Ballagadus rossi*) (*Smithson, Richards & Clack, 2016*). These tooth plates are wider (length:width < 1), spoon-shaped, and bear fewer, subparallel ridges (*Challands et al., 2019*). The 'modern' lungfish dental morphology (*sensu Schultze & Chorn (1997)*, *i.e.,* Mesozoic and Cenozoic lungfishes) is characterised by sharp ridges of entirely fused tooth cusps, and first appears, according to current knowledge, in the Visean stage in the genus *Sagenodus* (*Owen, 1867*; *Beeby, Smithson & Clack, 2020*). This highly successful morphotype is still observed in the extant *Neoceratodus forsteri* and has been predominant in this group for about 320 million years (*Smithson, Richards & Clack, 2016*).

Lungfish evolution across the Devonian-Carboniferous boundary is a point of contention (*Schultze & Chorn, 1997*; *Ahlberg, Smith & Johanson, 2006*; *Sallan & Coates, 2010*; *Lloyd,*

*Wang & Brusatte, 2012*; *Pardo, Huttenlocker & Small, 2014*; *Kemp, Cavin & Guinot, 2017*). While the aftermath of the Hangenberg extinction event was a time of great diversification for actinopterygians and chondrichthyans, sarcopterygian fishes saw a steep decline in abundance and diversity in the Early Carboniferous (*McGhee, 1996*; *McGhee et al., 2004*; *McGhee et al., 2013*; *Sallan & Coates, 2010*; *Friedman & Sallan, 2012*; *Greif, Ferrón & Klug, 2022*). Until recently, it was assumed that this applied to lungfishes too. The known fossil record suggested that dipnoan diversity started decreasing in the Late Devonian, and later plummeted after the Hangenberg event (*Ahlberg, Smith & Johanson, 2006*; *Kemp, Cavin & Guinot, 2017*; *Challands et al., 2019*). Post-Devonian stem lineages were thought to be the few survivors of the end-Devonian mass extinction, and to constitute a distinct phylogenetic entity with little or no overlap with Devonian taxa (*Sallan & Coates, 2010*; *Lloyd, Wang & Brusatte, 2012*; *Kemp, Cavin & Guinot, 2017*). This division was mostly based on the apparent mutual exclusivity of Devonian and post-Devonian morphologies. The past decade of research has consistently challenged these ideas.

A number of key studies have brought new fossil and phylogenetic evidence to light, which suggest that the Devonian-Carboniferous boundary may have been more permeable for lungfishes than previously thought. Specifically, new material from the Famennian and Tournaisian of Greenland and the UK, respectively, have revealed surprisingly diverse lungfish assemblages (*Carpenter et al., 2014*; *Smithson, Richards & Clack, 2016*; *Challands et al., 2019*; *Clack et al., 2019*), contributing to blurring the so-called 'Romer's gap' at the beginning of the Carboniferous. In total, the authors describe six new genera and ten new species, most of them from the Tournaisian, as well as some undiagnosed material. The dental elements in particular display a range of morphologies. This relatively high diversity encountered right after the Hangenberg event implies that lungfishes were not as strongly affected by the extinction as other sarcopterygian groups (*Smithson, Richards & Clack, 2016*; *Challands et al., 2019*; *Clack et al., 2019*). Some lineages persisted into the Carboniferous and underwent a second smaller radiation, allowing lungfishes to swiftly recover (*Smithson, Richards & Clack, 2016*; *Challands & Den Blaauwen, 2017*; *Challands et al., 2019*; *Clack et al., 2019*). Moreover, morphological similarities between Famennian and Tournaisian taxa suggest a higher level of diversity in the latest Devonian, and a significant amount of mixture across the boundary (*Clack et al., 2019*; *Challands et al., 2019*). This is reflected in the phylogenetic analyses that take this new material into account. The results yield low clade support for a monophyletic Carboniferous group and find that Famennian and Mississippian taxa consistently cluster together (*Challands et al., 2019*). Not only does this invalidate the previously accepted divide, but it also implies that many derived lungfish lineages find their roots deep in the Devonian, rather than originating after the Hangenberg event (*Clack et al., 2019*). Until now, the only potential fossil evidence for this was a pair of undescribed tooth plates from the Frasnian of Russia, which apparently share similarities with those of the Carboniferous genus *Ctenodus* (*Challands et al., 2017*).

Here we present a new tooth plate from the Famennian layers of the Tafilalt region in the Moroccan Anti-Atlas. We describe the specimen's highly derived features, which are unseen in any other Devonian taxa, and discuss its implications for our understanding of lungfish evolution across the Devonian-Carboniferous boundary. This specimen also adds

to our very limited knowledge of dipnoans and other osteichthyans from the Devonian of Morocco, and the African continent.

## MATERIALS & METHODS

### Material

PIMUZ A/I 5339 was collected in the Tafilalt region, Southwest of the city of Merzouga in the Moroccan Anti-Atlas (31.03883°N, 4.10194°W, Figs. 1A, 1B). A permit for the collection and export of fossil material was granted by the Ministère de l'Énergie, des Mines et de l'Environnement, Rabat, Morocco (permit N° 1571/DE/DG). It was found in the Upper Famennian *Gonioclymenia* limestone layer (363.6–362 ma, *Becker et al., 2020*) of Rich Tamirant, a site located on the eastern end of the Amessoui Syncline (*Hollard, 1981a*; *Hollard, 1981b*; *Klug & Pohle, 2018*). The Rich Tamirant locality is part of the Tafilalt Platform, located in the Northern coast of Gondwana in the Famennian (*Kocsis & Scotese, 2021*) (Fig. 1C). This platform was submerged from the Silurian to the Early Carboniferous and separates the Maider and Tafilalt epicontinental basins (*Wendt, 1985*; *Wendt, 1988*; *Wendt, 2021a*; *Wendt, 2021b*). The Devonian layers of this platform are incredibly rich in invertebrate and vertebrate remains (*Klug & Pohle, 2018*; *Frey et al., 2018*; *Frey et al., 2020*). However, osteichthyan material remains scarce in the Tafilalt region (*Lelièvre & Janvier, 1986*; *Lelièvre & Janvier, 1988*; *Aquesbi, 1988*; *Campbell et al., 2002*; *Rücklin & Clément, 2017*). PIMUZ A/I 5339 represents only the third instance of a Devonian lungfish from Morocco, and the fourth on the African continent: *Isityumzi mlomomde* from the Famennian of South Africa (*Gess & Clement, 2019*), *Dipnotuberculus gnathodus* from the Givetian of Morocco (*Campbell et al., 2002*), and an undescribed palate from the Frasnian of Morocco likely pertaining to *Dipnotuberculus* (*Murray, 2000*; *Gess & Clement, 2019*).

### Associated microfossils

Chemical preparation of the sediment directly attached to PIMUZ A/I 5339 revealed conodont dental elements which we identify as *Palmatolepis* cf. *gracilis*, confirming this material is of Upper Famennian origin (Figs. 2A–2C). Our sample also includes a chondrichthyan tooth, which we identified as *Denaea* cf. *fournieri* (Fig. 2D) (*Pruvost, 1922*; *Ginter & Hansen, 2010*; *Ginter, Hampe & Duffin, 2010*; *Ginter et al., 2015*).

### Preparation and photography

We chemically prepared the specimen using a 10% acetic acid solution. We soaked it in an acid bath for three to four days, then washed it with distilled water and brushed off the residue, which was sieved for microfossils. After each wash, we coated the tooth plate with a light layer of paraffin wax. We repeated these steps until the sediment matrix was dissolved from the specimen. Finally, we mechanically removed any additional residue using air scribes. We photographed the prepared specimen using a Keyence VHX-7000 digital microscope.

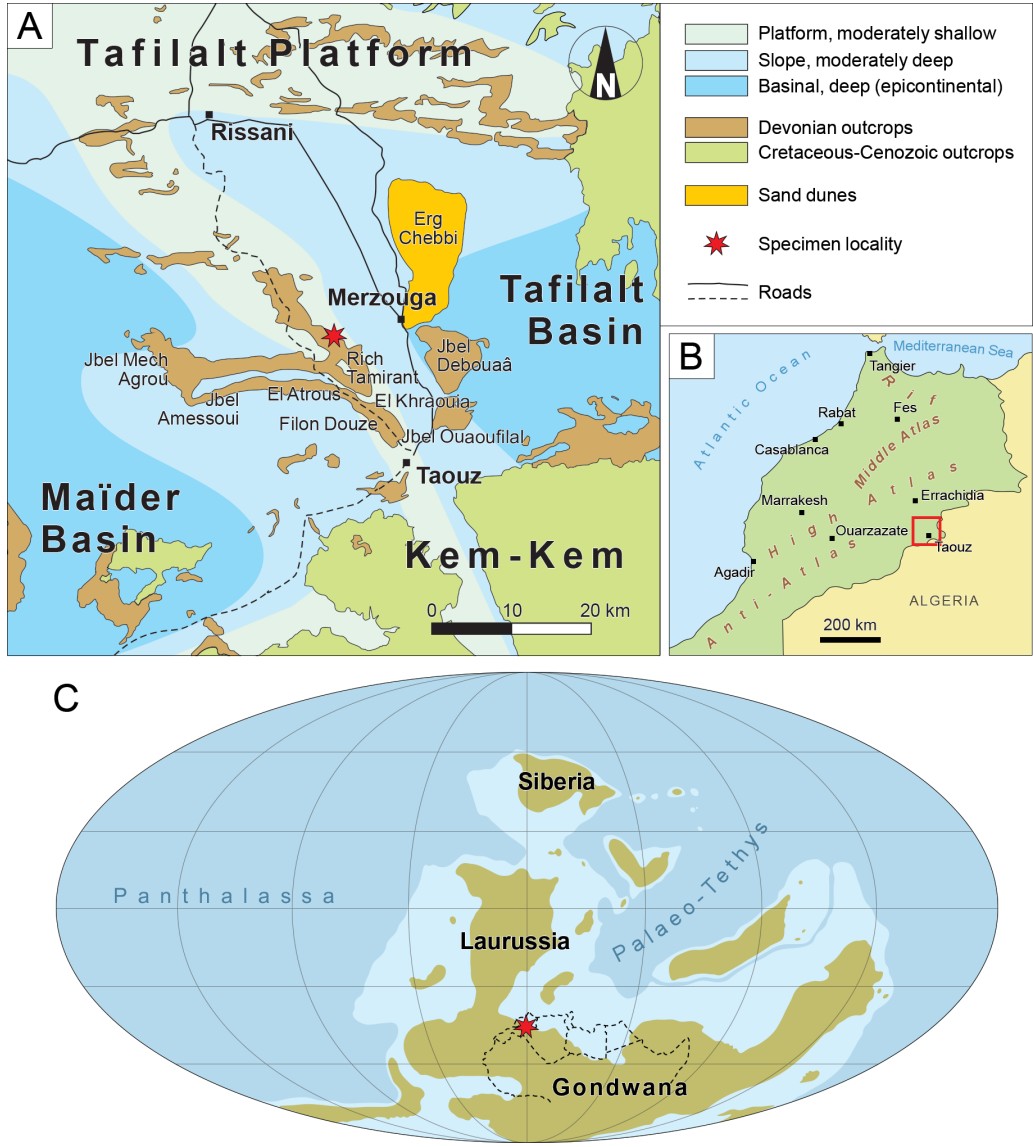

**Figure 1 Geographical and palaeogeographical context of PIMUZ A/I 5339.** (A) Map of the Tafilalt region and geographical location of the fossil locality. (B) Map of Morocco. (C) Palaeogeographical world map of the major landmasses during the Famennian (after *Kocsis & Scotese, 2021*). Red square in B indicates the region shown in A. Red star in A and C indicates the fossil locality. Dotted lines in C show the modern outlines of North Africa.

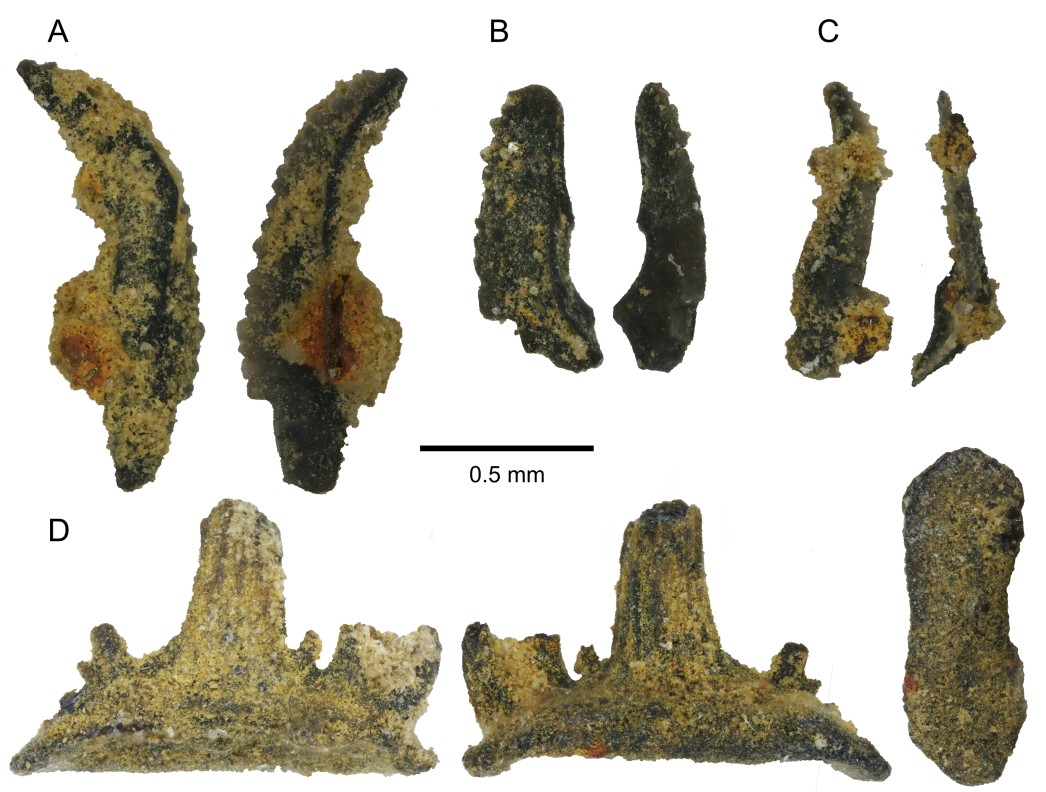

**Figure 2** **Conodont and chondrichhtyan material associated with PIMUZ A/I 5339.** (A–C) *Palmatolepis* cf. *gracilis* conodont dental elements in lateral views. (D) Chondrichthyan tooth in lingual, labial and basal views (left-to-right).

## RESULTS

### Systematic palaeontology

Superclass OSTEICHTHYES *Huxley, 1880*
Class SARCOPTERYGII *Romer, 1955*
Order DIPNOI *Müller, 1845*
Family ?SAGENODONTIDAE *Jaekel, 1911*
gen. indet.
(Fig. 3)

**Material**–PIMUZ A/I 5339, an isolated tooth plate.
**Locality**–Rich Tamirant, Tafilalt Platform, Eastern Anti-Atlas, Morocco.
**Stratigraphy**–Dra Group, Lemgaïrinat Formation, *Gonioclymenia* limestone, Famennian UD V-B, Late Devonian.
**Remarks**–Given PIMUZ A/I 5339's striking similarity to the Carboniferous genus *Sagenodus* (*Schultze & Chorn, 1997*; *Olive, Clément & Pouillon, 2012*; *Beeby, Smithson & Clack, 2020*), we tentatively refer this specimen to the family Sagenodontidae. Taxonomic

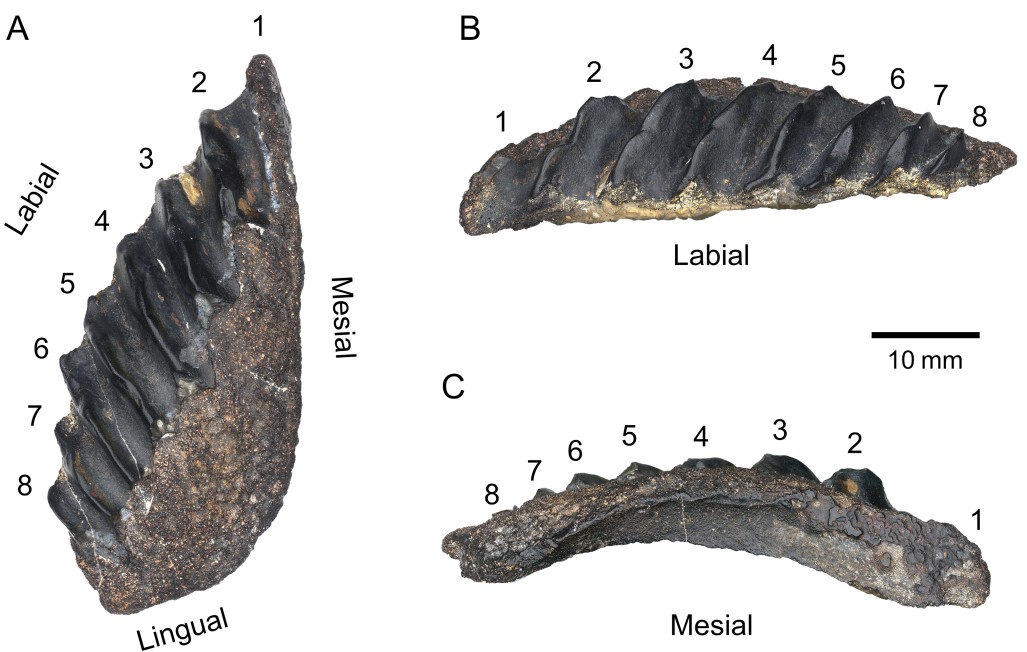

**Figure 3** **PIMUZ A/I 5339 dipnoan tooth plate.** (A) Occlusal view. (B) Labial view. (C) Mesial view. Individual tooth rows are numbered 1–8.

identification based solely on isolated tooth plates can be tenuous due to the significant amount of intraspecific variation seen in these structures (*Reed, 1985*; *Kemp, 1997*; *Kemp, 2005*; *Friedman, 2007*). Therefore, PIMUZ A/I 5339 cannot be undisputedly attributed to family level in the absence of associated cranial material (*Clack et al., 2019*; *Mondéjar-Fernández et al., 2020*). Furthermore, as the tooth plate is dissociated from any jaw material, it cannot be determined whether it is a right upper (pterygoid) or left lower (pre-articular) tooth plate without a genus-level identification. Note that in some aspects, PIMUZ A/I 5339 fits the characters used by *Schultze & Chorn (1997)* to distinguish the lower tooth plates of *Sagenodus copeanus* from the upper ones. However, this is not sufficient to discern the nature of this plate with certainty.

## Description

The tooth plate is broad, convex, and subtriangular in shape, reaching a maximum length of approximately 60 mm and a width of 25 mm, giving it a 2.4 length-width ratio. The labial margin is flat in lateral view and convex in occlusal view. The labio-lingual junction is acute and relatively shallow. The lingual margin is short (approx. 15 mm) and straight in occlusal view but forms a deep convex arch in lateral view ending in a rounded medio-lingual junction. The mesial margin is straight in occlusal view and convex in mesial view and gets gradually broader anteriorly. The occlusal surface is highly damaged: the petrodentine layer on the medio-lingual half of the plate, as well as the entirety of the first tooth ridge, is broken off along a straight line, exposing the underlying bone and hollow pulp cavities. The tooth plate counts eight straight rows of teeth fused into blade-like ridges with deep rounded

furrows. The rows do not radiate from a single point, but from a broad medio-lingual origin. The maximum tooth ridge angle, measured between the first and eighth ridge, is approximately 33.5°. The angle between adjacent tooth ridges is highest between the first and second ridges, and decreases posteriorly, where the ridges are subparallel to each other. Due to the damage on the proximal half of the plate, these angles cannot be precisely measured. Individual denticles on each row are not recognisable. It is unclear whether this lack of clear denticulation, as seen in *Sagenodus quinquecostatus* (*Beeby, Smithson & Clack, 2020*) is a diagnostic feature of this specimen, or if it is due to wear during life or after death. The presence of irregular bumps on the distal part of the ridges suggests the tooth plate may have borne conspicuous tooth cusps which eventually wore off. The size of the plate, the high degree of fusion, and the erosion of teeth suggest this individual was an adult.

## DISCUSSION

### Taxonomic affinity

PIMUZ A/I 5339 consists solely of an isolated tooth plate (Fig. 3). The specimen has highly derived features that clearly distinguish it from any other known lungfish from the Devonian and the lowermost Carboniferous (Tournaisian) (Fig. 4). In the vast majority of Devonian tooth plated dentitions, the surface of the plate is covered with individual, unfused, and rounded tooth cusps. In more derived forms from the Famennian (*Challands et al., 2019*) such as *Andreyevichthys epitomus* (*Krupina, 1987*) or *Adelargo schultzei* (*Johanson & Ritchie, 2000*), tooth plates are roughly triangular in shape, and tooth cusps are aligned in radially arranged rows, which converge at the mediolingual edge of the plate (Figs. 4A, 4B). These rows may also display interrow denticles (*Smith & Krupina, 2001*; *Mondéjar-Fernández et al., 2020*). PIMUZ A/I 5339 is the only known example of a Devonian tooth plate with entirely fused tooth cusps, and sharp, continuous tooth rows (Fig. 3). Some Tournaisian taxa do display partial fusion of tooth cusps, like *Occludus romeri* (Fig. 4E) (*Smithson, Richards & Clack, 2016*), but not nearly to the same extent as in PIMUZ A/I 5339. In addition, the characteristic subtriangular shape of this specimen and its relatively low number of tooth ridges are not seen in any Tournaisian taxon either. It is not until the Visean stage that we see tooth plates of a shape and size comparable to PIMUZ A/I 5339 with sharp and entirely fused tooth rows in *Sagenodus quinquecostatus* (Figs. 4C, 4F) (*Beeby, Smithson & Clack, 2020*). As such, we believe PIMUZ A/I 5339's morphology justifies its placement within the Sagenodontidae.

It is difficult to evaluate the specimen's affinity by comparing it to other sagenodontids, as there is no consensus on the taxa which make up this family other than *Sagenodus*. Genera which are customarily attributed to the Sagenodontidae, or likened to *Sagenodus*, include *Straitonia* and *Parasagenodus* from the Mississippian (*Thomson, 1965*; *Vorob'yeva, 1972*; *Sharp & Clack, 2012*), *Megapleuron* from the Pennsylvanian and Lower Permian (*Gaudry, 1881*; *Schultze, 1977*), and *Aphelodus* from the Triassic (*Kemp, 1993*). However, re-evaluations of the fossil material and new phylogenetic analyses have raised uncertainty about their relatedness to Sagenodontidae (*Campbell & Barwick, 1990*; *Lloyd, Wang &*

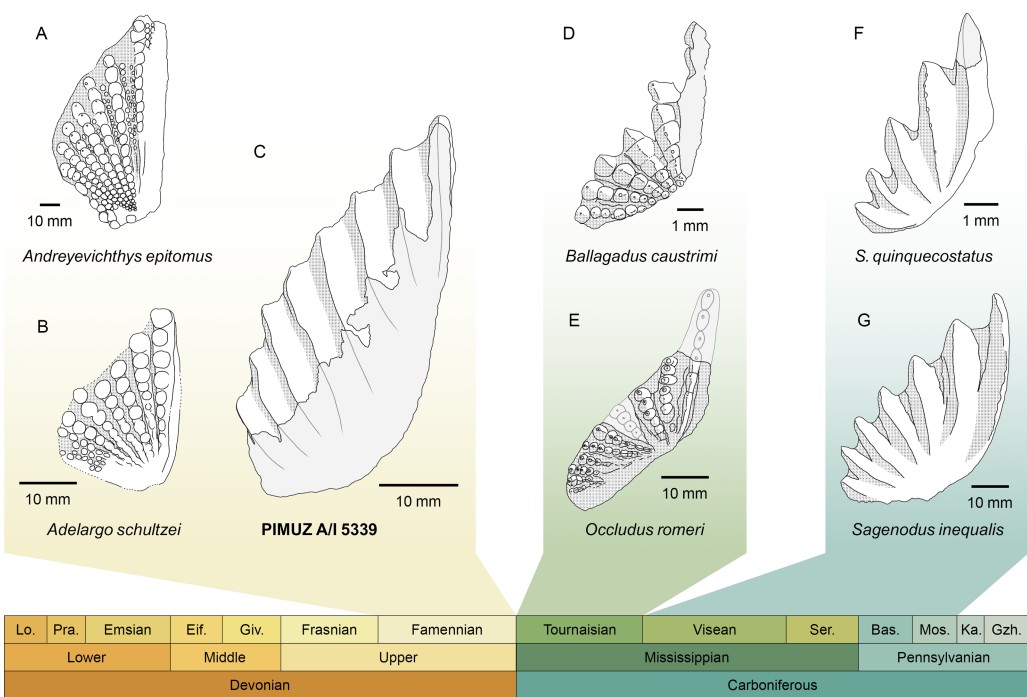

**Figure 4** **Comparison of PIMUZ A/I 5339 to other Devonian and Carboniferous dipnoan pterygoid tooth plates.** (A) *Andreyevichthys epitomus* (reversed), from *Smith & Krupina (2001)*. (B) *Adelargo schultzei*, from *Johanson & Ritchie (2000)*. (C) PIMUZ A/I 5339. (D) *Ballagadus caustrimi* , from *Smithson, Richards & Clack (2016)*. (E) *Occludus romeri* (reversed), from *Smithson, Richards & Clack (2016)*. (F) *Sagenodus quinquecostatus* (reversed), from *Beeby, Smithson & Clack (2020)*. (G) *Sagenodus inequalis* (reversed), from *Beeby, Smithson & Clack (2020)*. Abbreviations of stages: Lo, Lochkovian; Pra, Pragian; Eif, Eifelian; Giv, Givetian; Ser, Serpukhovian; Bas, Bashkirian; Mos, Moscovian; Ka, Kasimovian; Gzh, Gzhelian.

*Brusatte, 2012*; *Kemp, Cavin & Guinot, 2017*; *Challands et al., 2019*; *Clack et al., 2019*). These, and a number of other genera regarded as 'transitional' forms (*e.g.*, *Ctenodus*, *Conchopoma*, *Uronemus*, *Gnathorhiza*, *Palaeopichthys*, *Celsiodon*, *Persephonichthys*) (*Pardo, Huttenlocker & Small, 2014*; *Kemp, Cavin & Guinot, 2017*; *Challands et al., 2019*; *Clack et al., 2019*), mark the end of 'primitive' Palaeozoic lungfishes and the origin of 'modern' lineages. Due to this unique position in dipnoan phylogeny, 'transitional' forms often exhibit a mix of ancestral and derived traits, making their systematic assignment and phylogenetic placement highly liable to vary depending on the analysis. As such, we chose to limit our comparison of PIMUZ A/I 5339 to *Sagenodus*, and base our tentative diagnosis on the striking similarity of their dental morphology. Making a lower-rank identification would require further cranial material, especially elements of the skull roof (*Beeby, Smithson & Clack, 2020*).

Lungfish dental elements can be indicators of taxonomic affinity, but they are not reliable for precise identification efforts (*Mondéjar-Fernández et al., 2020*). Firstly, a single, incomplete, and isolated tooth plate does not hold enough morphological information to obtain a precise phylogenetic placement. For this, additional skeletal remains are necessary,

especially in such a time of exceptionally high species diversity (*Clack et al., 2019*). Secondly, dipnoan tooth plates also show great variation due to ontogenetic variability, especially in the Devonian, when developmental plasticity is believed to have been relatively high (*Reed, 1985*; *Ahlberg, Smith & Johanson, 2006*). Developmental change, genetic and pathological anomalies, wear and modification throughout an individual's lifetime can also hamper the identification of isolated tooth plates (*Kemp, 1997*; *Kemp, 2005*; *Kemp, Cavin & Guinot, 2017*; *Challands et al., 2019*). Only when fossil ontogenetic series are well documented can we build a comprehensive morphometric framework and attempt to diagnose taxa based on their tooth plates alone (*Smithson, Richards & Clack, 2016*; *Mondéjar-Fernández et al., 2020*). Finally, post-burial deformation can also alter some aspects of tooth plate morphology, which thus renders it unreliable in phylogenetic studies (*Kemp, 1977*; *Kemp, Cavin & Guinot, 2017*; *Challands et al., 2019*).

## The origin of the 'modern' lungfish tooth plate

The earliest unequivocal record of dipnoan tooth plates with derived crown-like features—other than PIMUZ A/I 5339—is that of *Sagenodus quinquecostatus* from the Visean and Serpukhovian of Scotland (*Beeby, Smithson & Clack, 2020*). This 'Sagenodontid' dental configuration with sharp ridges of fused teeth has been traditionally described as representing the 'modern' lungfish dental morphology (*sensu Schultze & Chorn, 1997*). It is partly for this reason that *Sagenodus* is regarded as the Palaeozoic dipnoan closest to extant lungfishes (*Schultze & Chorn, 1997*). However, PIMUZ A/I 5339 shows that this morphology had already evolved by the Famennian at the latest, pushing back the origin of 'modern' lungfish dentition.

Assuming a close phylogenetic affinity to Sagenodontidae, the discovery of PIMUZ A/I 5339 in North Africa may also have palaeogeographical implications for this lineage. The earliest sagenodontids (excluding PIMUZ A/I 5339), *Sagenodus quinquecostatus* and potentially *Straitonia waterstoni*, date back to the Visean of the UK (*Sharp & Clack, 2012*; *Beeby, Smithson & Clack, 2020*), an area of the world corresponding to the southern coast of Laurussia at the time (*Kocsis & Scotese, 2021*). Note that *Carpenter et al. (2014)* do suggest an earlier occurrence of *Sagenodus* in the Tournaisian of Scotland in the form of juvenile tooth plates, although this diagnosis has been put into question (*Beeby, Smithson & Clack, 2020*). Throughout the rest of the Palaeozoic, remains of *Sagenodus* and other presumed sagenodontids occur in North America, Europe and Russia (*Vorob'yeva, 1972*; *Schultze, 1977*; *Campbell & Barwick, 1990*; *Schultze & Chorn, 1997*; *Olive, Clément & Pouillon, 2012*; *Sharp & Clack, 2012*; *Beeby, Smithson & Clack, 2020*), suggesting westward and eastward dispersal in the Late Mississippian from a putative British point of origin. *Brownstein, Harrington & Near (2023)* estimate a European ancestral range for 'transitional' lungfish lineages including genera such as *Ctenodus*, *Conchopoma*, and *Sagenodus*. The Famennian occurrence of PIMUZ A/I 5339 suggests instead that the early evolution of sagenodontids, and the 'modern' lungfish tooth plate, may have occurred initially in Gondwana, along its northern coast (*e.g.*, North Africa), during the Late Devonian.

This change in distribution entails a major shift in ecology. While PIMUZ A/I 5339 inhabited a shallow marine platform in the Famennian (Fig. 1A) (*Wendt, 1985*; *Wendt,*

*1988*; *Wendt, 2021a*; *Wendt, 2021b*), its Mississippian relatives in Britain are associated with complex near-shore freshwater deposits with periodic inputs of seawater (*Jones, 2005*; *Carpenter et al., 2014*; *Bennett et al., 2017*; *Challands et al., 2019*). More generally, sarcopterygians in the Devonian occupied large homogeneous marine and lacustrine environments, which give way to more seasonal freshwater and brackish coastal floodplain environments in the Tournaisian (*Bennett et al., 2016*; *Kearsey et al., 2016*; *Millward et al., 2018*; *Challands et al., 2019*). Derived *Sagenodus*-like tooth plates like PIMUZ A/I 5339 may be associated with a generalist feeding strategy (*Smithson, Richards & Clack, 2016*), which would have allowed them to survive these ecological changes, quickly adapt, and thrive in new conditions. This inference is based on the similarity of this dental morphotype to the dentition of the extant lungfish *Neoceratodus forsteri* (*Smithson, Richards & Clack, 2016*), which is known to feed on plant material, small invertebrates, small fish, and tadpoles (*Kemp, 1986*; *Kind, 2010*). Furthermore, the fish-bearing coastal floodplain ecosystems of the Tournaisian preserve a rich assortment of small plant, invertebrate, and vertebrate fossils (*Bennett et al., 2017*; *Challands et al., 2019*; *Otoo et al., 2019*). However, while its tooth plates do allow it to process a wide range of items, *Neoceratodus forsteri* is a slow-moving organism with relatively weak jaws, restraining its effective feeding ability (M Coates, 2024, pers. comm.). Conversely, its skeleton comprises much more cartilage than *Sagenodus* and other fossil taxa from this time, making their biomechanical properties significantly different. Therefore, while dental morphology can partially inform on diet, it does not represent a one-to-one association with feeding ecology, and comparison with extant taxa should be made cautiously. In any case, *Sagenodus*-like dentitions did presumably grant some level of versatility compared to other contemporary morphotypes. The occurrence of a *Sagenodus*-like tooth plate (PIMUZ A/I 5339) in the Late Devonian suggests that this derived generalist morphotype did not evolve in the Carboniferous as a response to new environmental pressures, but rather that it was exapted to survive and thrive in the wake of the Hangenberg crisis.

### Dipnoan evolution across the Devonian-Carboniferous boundary

The discovery of PIMUZ A/I 5339 provides strong support for recent hypotheses about dipnoan morphological evolution across the Devonian-Carboniferous boundary (*Smithson, Richards & Clack, 2016*; *Challands et al., 2019*; *Clack et al., 2019*). It presents the first unambiguous and direct fossil evidence that a 'modern' tooth plate morphology, which was thought to be restricted to post-Devonian times, had in fact already evolved in the Devonian. Its striking resemblance to *Sagenodus* dentitions suggests a close phylogenetic relationship (Fig. 4), implying that the new dipnoan represented by PIMUZ A/I 5339 may have an affinity within Carboniferous lungfishes. Unfortunately, its exact position in the dipnoan tree cannot yet be determined due to the absence of further material (*Clack et al., 2019*; *Mondéjar-Fernández et al., 2020*). Yet, the possibility of convergent evolution must also be considered. However, applying the laws of parsimony, we posit that it is unlikely that this unique and massively successful morphology was convergently evolved by another, more stem-ward lineage, which did not survive the Hangenberg extinction event. Thus, we can confidently state that PIMUZ A/I 5339 and *Sagenodus* are probably closely related.

The morphology and presumptive systematic attribution of PIMUZ A/I 5339 to the Sagenodontidae provides further support for a permeable Devonian-Carboniferous boundary for lungfishes with a significant amount of overlap between Famennian and Carboniferous taxa (*Smithson, Richards & Clack, 2016*; *Challands et al., 2019*; *Clack et al., 2019*). It also confirms that many post-Devonian groups, like the Sagenodontidae in this case, may actually be deeply rooted in the Devonian (*Smithson, Richards & Clack, 2016*; *Challands et al., 2019*; *Clack et al., 2019*). Note that this overlap in dental morphologies is mostly recognisable in Famennian lungfish faunas, but much less in the Frasnian. Tooth plates from this stage are morphologically similar to Famennian and Carboniferous forms, bearing organised rows of unfused or partially fused tooth cusps (*e.g.*, *Gogodipterus*, *Scaumenacia*, *Rhinodipterus*) (*Traquair, 1893*; *Gross, 1956*; *Long, 1992*), but the overall landscape of dipnoan dental diversity is significantly different. In the Famennian and Carboniferous, tooth plates are the most common type of dentition, while dentine plates are entirely absent from the record, and denticulated plates remain in a few taxa and disappear early in the Mississippian (*Lloyd, Wang & Brusatte, 2012*; *Smithson, Richards & Clack, 2016*; *Clack et al., 2019*; *Lebedev, Krupina & Linkevich, 2019*). In the Frasnian, all three types of dentitions are roughly equally distributed across taxa, with a slight overrepresentation of denticulated plates (*Lloyd, Wang & Brusatte, 2012*). This represents a major change in lungfish dental diversity at the Frasnian-Famennian boundary, while Famennian diversity patterns seem to remain relatively intact in the Tournaisian (*Smithson, Richards & Clack, 2016*; *Challands et al., 2019*). This suggests that the Kellwasser event may have had a greater impact on lungfishes than the Hangenberg crisis, thus shaping the diversity of Famennian and post-Devonian faunas.

Still, in comparison with other vertebrate groups, lungfishes were among those which were left somewhat unscathed by the end-Devonian extinction (*Smithson, Richards & Clack, 2016*; *Challands et al., 2019*). While 'placoderms' went extinct and most sarcopterygians suffered greatly, dipnoans, actinopterygians, and chondrichthyans were much less affected, establishing a faunal composition that is mostly conserved up to present day (*McGhee, 1996*; *McGhee et al., 2004*; *McGhee et al., 2013*; *Sallan & Coates, 2010*; *Friedman & Sallan, 2012*; *Greif, Ferrón & Klug, 2022*). While the latter two radiated in the Early Carboniferous, lungfishes gradually diminished in diversity as the successful *Ctenodus* and *Sagenodus* dental morphotypes became widely established (*Clack, Sharp & Long, 2010*; *Smithson, Richards & Clack, 2016*). PIMUZ A/I 5339 represents a key addition to our changing understanding of dipnoan and sarcopterygian evolution at the Devonian-Carboniferous boundary from the African Palaeozoic fossil record.

## CONCLUSIONS

PIMUZ A/I 5339 represents the first fossil evidence for the Devonian origin of a Carboniferous lungfish lineage. It corroborates the idea that the effect of the Hangenberg event on lungfishes was not as a dramatic as it was on other marine vertebrates. The Devonian-Carboniferous boundary appears to have been quite permeable for this group, with the occurrence of crownward taxa in the Devonian, and of stemward survivors in the

Carboniferous. Furthermore, the Gondwanan PIMUZ A/I 5339 bears a striking resemblance to *Sagenodus*, a genus appearing in the Visean of Laurussia and often considered to be the first representative of the 'modern' lungfish body plan. A tooth plate in isolation cannot support a confident species diagnosis in lungfishes, but we are confident that PIMUZ A/I 5339 is closely related to *Sagenodus*, thus pushing back the origin of the 'modern' lungfish dentition to the Famennian. With the availability of further material, future studies could incorporate this new data into phylogenetic and morphometric studies to determine the precise phylogenetic affinity of this taxon. Furthermore, micro-CT scanning would enable a non-destructive a histological study of PIMUZ A/I 5339. Comparing it to post-Devonian taxa with similar dental morphologies, such as *Sagenodus*, would allow us to investigate the changes in dental development of this successful and widespread tooth plate morphology through time.

## ACKNOWLEDGEMENTS

We thank Mohammed Mezane (Merzouga, Morocco) who discovered PIMUZ A/I 5339, Dr. Marc Leu (Department of Palaeontology, University of Zurich, Switzerland) for helping with the identification of conodont material, Prof. Michael Coates (Department of Organismal Biology and Anatomy, University of Chigaco, USA) and Dr. Martin Brazeau (Department of Life Sciences, Imperial College London, UK) for their constructive suggestions and advice.

### Funding

This research was funded by the Swiss National Science Foundation project nr. 205320_215642. Jorge Mondéjar Fernández was supported by the Louis Gentil-Jacques Bourcart prize of the French Academy of Sciences. Alice M. Clement was supported by the Australian Research Council (DP 220100825). The funders had no role in study design, data collection and analysis, decision to publish, or preparation of the manuscript.

### Grant Disclosures

The following grant information was disclosed by the authors:
Swiss National Science Foundation: 205320_215642.
Louis Gentil-Jacques Bourcart prize of the French Academy of Sciences.
Australian Research Council: DP 220100825.

### Competing Interests

The authors declare there are no competing interests.

### Author Contributions

- Amin El Fassi El Fehri conceived and designed the experiments, performed the experiments, analyzed the data, prepared figures and/or tables, authored or reviewed drafts of the article, and approved the final draft.

- Alice M. Clement analyzed the data, authored or reviewed drafts of the article, and approved the final draft.
- Jorge Mondéjar Fernández analyzed the data, authored or reviewed drafts of the article, and approved the final draft.
- Merle Greif performed the experiments, analyzed the data, prepared figures and/or tables, and approved the final draft.
- Christian Klug conceived and designed the experiments, analyzed the data, authored or reviewed drafts of the article, and approved the final draft.

### Field Study Permissions

The following information was supplied relating to field study approvals (i.e., approving body and any reference numbers):

Fieldwork in Morocco, fossil collection, and export to Switzerland was approved by the Ministère de l'Énergie, des Mines et de l'Environnement, Rabat, Morocco (permit number: 1571/DE/DG).

### Data Availability

PIMUZ A/I 5339 is reposited at the UZH DoP Collection Database, Department of Paleontology (DoP), University of Zurich (UZH), Karl-Schmid-Strasse 4, 8006 Zurich, Switzerland; MySQL-ID: 57500'

https://www.pim.uzh.ch/en/sammlung/db.html.

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
