# Peer review of "A new origin of the ‘modern’ lungfish dentition revealed by taxonomic overlap between Devonian and Carboniferous dipnoans"

_PeerJ, doi:10.7717/peerj.19389_

## Round 0.1 · original submission · Major Revisions

I have read the paper and the reviews and agree that this paper needs a major revision. Please read carefully all comments provided by the three reviewers.

·

Basic reporting

This manuscript reports an excellent discovery, a lungfish fused tooth plate of putative Sagenodus-type from the Devonian of Morocco. This is an interesting finding, and potentially pushes the origin of modern dipnoan dentition closer to the origin of the lungfish clade.

The manuscript's English is professional and of acceptable quality.

Experimental design

The experimental design here is good, but more information is needed. First, was the sediment that yielded the conodont Palmatolepis directly attached to the tooth plate? Or was it from a sediment sample taken from nearby?

Second, even before the authors mentioned it in line 383, it seemed clear to me that a CT scan is required to get a better sense of the interior structure of this fossil. It is curious that PIMUZ A/I 5339 shows no sign of denticulation along the tooth ridges, as seen for example in Sagenodus quinquecostatus. Lack of such denticulation might be considered a more, rather than a less, derived character state. This has implications for understanding the sudden appearance versus the stepwise evolution of the modern, fused tooth plate type.

Validity of the findings

I appreciate the authors' caution in terms of taxonomic assignment. I agree that the specimen should be placed in the Sagenodontidae with question. It seems likely, though, that PIMUZ A/I 5339 belongs to a previously undescribed genus. The authors should consider naming a new genus and species based on this specimen after CT results become available.

I agree that the fossil is from an adult animal.

Additional comments

This manuscript really needs to add CT data before moving forward. I am familiar with equipment at Boston College that would be ideal for analysis of a specimen this size. A good scan could be made quickly and easily, and would very likely be very informative.

Comments on text:

Line 1: regard not regards

Line 423: dipnoan not Dipnoan

·

Basic reporting

There are no problems in any of these areas.

Experimental design

There are no problems in any of these areas. The article is descriptive, not experimental.

Validity of the findings

A bit marginal. Yes, this is clearly a Late Devonian lungfish tooth plate resembling the Carboniferous genus Sagenodus, and that's quite an interesting discovery, but it is just a single isolated data point and I really feel the authors expand the discussion of its possible significance beyond what it really merits. I especially don't like the section about biogeography and dispersal (lines 266-283), where this single specimen is taken as evidence that the Sagenodus-type dentition originated in North Africa! This is not even remotely justified, you would need much stronger evidence from multiple specimens to draw such a conclusion. The discovery of the specimen is compatible with a North African origin for the lineage, but that's not the same thing as meaningful positive evidence.

Additional comments

I disagree with the characterisation of toothplate evolution in early lungfishes given on lines 62-75. Diabolepis and Dipnotuberculus have completely different dentitions. In Diabolepis the discrete teeth are for the most part organised in neat rows; Dipnotuberculus on the other hand has no teeth at all, only lumps and bumps of dentine that show no obvious trace of radial arrangment. The claim (lines 64-65) that teeth organised into rows emerge as a character state in the Late Devonian, is false. Radiating tooth rows are already present in Diabolepis (Early Devonian), Speonesydrion (Early Devonian), Tarachomylax (Early Devonian), Dipterus (Middle Devonian) and several more taxa besides. I also think it's a bit of a stretch to consider the Sagenodus morphotype as 'modern' and directly comparable with those of extant lungfishes. The tooth plates of Neoceratodus are quite similar to similar to Sagenodus, but those of Protopterus and Lepidosiren are really rather different with just three knife-like blades radiating from a single point of origin.

Reviewer 3 ·

Basic reporting

No comment. This paper is an excellent description of new lungfish material from an important interval in the study of Paleozoic fish evolution. I have no complaints about the core of the paper but find the small section discussing an associated shark tooth fossil to be extraneous unless the authors can narrow its identification.

Experimental design

No comment. This paper follows the norms of description of new fossil material, especially if the material is limited but still diagnostic to some level.

Validity of the findings

No comment. My only complaint is that I feel the identity of the associated chondrichthyan tooth can be better resolved. Admittedly, I feel the discussion of the tooth does not add much to this paper especially as they have currently written the paper, and if they excluded it, it would not harm the paper. If they can affirm the tooth is referable to Denaea it would certainly make their position stronger (see notes below).

Additional comments

Line 29. Replace 'sagenodontid family' with 'the Sagenodontidae' or 'sagenodontids'.

Line 42-46. Consider combining sentences, "To explore lungfish evolutionary trends during this time, their dentitions provide excellent study material as they are robust, compact, and highly mineralized, which means they fossilize often and well (continue citations from line 43)."

Line 55. Replace dental diversity with dental disparity as you are acknowledging morphological disparity, not phylogenetic diversity.

Line 58-60. I know what you mean here, but please clarify the language as to whether some of these genera which extend across into the upper Carboniferous and beyond -- do they have this morphology or not? Readers may become confused.

Line 76. "has long been a topic of debate" -> "is contentious among researchers."

Line 130. Mya => Ma.

Line 147. This is comfortably referrable to Denaea, I see no characters that would support placing it within Stethacanthulus. I would go so far as to suggest you can find character support to suggest it is Denaea aff. fournieri Pruvost, 1922.

Line 151. Was the water distilled or otherwise purified?

Line 171-172. The sentence “This tentative placement is made while remaining conservative...” feels overwritten and unnecessary. The reader would be fine with the preceding sentence alone, by simply rewriting, “... we tentatively refer this specimen to the family Sagenodontidae.”

Line 365. “Denea” is a typo, it should be spelled “Denaea.” Sufficient characters exist to support a referral to Denaea aff. fournieri, which is well known across the time interval of the material being studied, unlike Stethacanthulus. I am not aware of a pre-middle Mississippian occurrence of Stethacanthulus. The authors should look at the discussions of characters of this species in Ginter and Hansen 2010, Ivanov 1999, etc.

---

## Round 0.2 · Minor Revisions

Thank you very much for your thorough revision. The reviewer is quite happy with it. There is one more small change requested by the reviewer that states, "I could ask for one final change; it would be in relation to lines 271-276 of the markup manuscript, where the text still gives the impression that pre-Famennian lungfishes have dentitions with a less regular arrangement of teeth. I don't know where this strange notion comes from, but it simply isn't true; it would be good if this sentence could be deleted". Please proceed to attend to this minor change.

·

Basic reporting

No problems.

Experimental design

No problems.

Validity of the findings

No problems.

Additional comments

This revision is significantly improved compared to the original and I am happy to recommend that it be accepted for publication. I have ticked 'accept', but if I could ask for one final change it would be in relation to lines 271-276 of the markup manuscript, where the text still gives the impression that pre-Famennian lungfishes have dentitions with a less regular arrangement of teeth. I don't know where this strange notion comes from, but it simply isn't true; it would be good if this sentence could be deleted.

---

## Round 0.3 · accepted · Accept

The authors have addressed all of the reviewers' comments and the manuscript is ready to be accepted for publication.